# Measuring and Reducing Model Update Regression in Structured Prediction for NLP

**Deng Cai**[*]
The Chinese University of Hong Kong
thisisjcykcd@gmail.com

**Elman Mansimov**
Amazon AWS AI Labs
mansimov@amazon.com

**Yi-An Lai**
Amazon AWS AI Labs
yianl@amazon.com

**Yixuan Su**[*]
University of Cambridge
ys484@cam.ac.uk

**Lei Shu**
Amazon AWS AI Labs
leishu@amazon.com

**Yi Zhang**
Amazon AWS AI Labs
yizhngn@amazon.com

## Abstract

Recent advance in deep learning has led to the rapid adoption of machine learning-based NLP models in a wide range of applications. Despite the continuous gain in accuracy, backward compatibility is also an important aspect for industrial applications, yet it received little research attention. Backward compatibility requires that the new model does not regress on cases that were correctly handled by its predecessor. This work studies model update regression in structured prediction tasks. We choose syntactic dependency parsing and conversational semantic parsing as representative examples of structured prediction tasks in NLP. First, we measure and analyze model update regression in different model update settings. Next, we explore and benchmark existing techniques for reducing model update regression including model ensemble and knowledge distillation. We further propose a simple and effective method, Backward-Congruent Re-ranking (BCR), by taking into account the characteristics of structured prediction. Experiments show that BCR can better mitigate model update regression than model ensemble and knowledge distillation approaches.

## 1 Introduction

*Model update regression* refers to the deterioration of performance in some test cases after a model update, even if the new model has on average better performance than the old model. For example, on a fixed test set, the new model may achieve 1% absolute improvement in accuracy. However, it does not necessarily mean that the new model gets 1% extra test cases correct. Instead, it may correct 6% mistakes made by its predecessor but also introduce 5% new errors. The occasional worse behavior of the new model can overshadow the benefits of overall performance gains and hinder its adoption. Imagine a newly updated virtual assistant stops understanding a user's favorite ways to inquire about the local traffic. This can severely degrade the user experience even if the assistant has been improved in other aspects.

Prior research (Shen et al., 2020; Yan et al., 2021; Xie et al., 2021) has investigated model update regression in *classification* problems of computer vision and natural language processing (NLP). However, the formalization of classification only has limited coverage of real-world NLP applications. For instance, the natural language understanding component behind modern virtual assistants aims to transform user utterances into semantic graphs, a task known as conversational semantic paring

---

[*]Most of the work was done during an internship at Amazon AWS AI Labs.

(Figure 1). In contrast to classification tasks, the global prediction of structured prediction (e.g., graph or tree) is often composed of many local predictions (e.g., nodes and edges). Some local predictions can be correct while the global prediction is imperfect. Therefore, model update regression can happen at fine-grained levels. Also, the output space is To our knowledge, the model update problem tasks.

In this work, we measure and analyze model update regression in two typical NLP structured prediction tasks: one general-purpose task (i.e., syntactic dependency parsing) and one application-oriented task (i.e., conversational semantic parsing). We set up evaluation protocols and test a range of model updates rooted in different aspects, including changes in model design, training data, and optimization process. Experiments show that model update regression is prevalent and evident across various model update settings.

The above finding shows a general and critical demand for techniques to reduce model update regression in structured prediction. Prior work for classification problems (Shen et al., 2020; Yan et al., 2021; Xie et al., 2021) has borrowed the idea of knowledge distillation (Hinton et al., 2015), where the new model (student) is trained to fit the output distribution of the old model (teacher). However, vanilla knowledge distillation (Hinton et al., 2015) cannot be directly applied to structured prediction models due to the intractability of the exact distribution of global prediction. Moreover, the goal prediction of structured prediction can be factorized in distinct ways. For example, a graph-based dependency parser (e.g., Dozat and Manning, 2017) scores every possible edge and searches for a maximum spanning tree, while a transition-based dependency parser (e.g., Ma et al., 2018) builds the tree incrementally through a series of actions. We refer to a model update between models with different factorizations as a *heterogeneous* model update. For heterogeneous model updates, existing factorization-specific approximation (e.g., Zmigrod et al., 2021a; Wang et al., 2021) or even knowledge distillation at the level of local predictions is inapplicable. To develop generic solutions that do not assume any specific factorizations, we instead resort to the sequence-level knowledge distillation (Kim and Rush, 2016), which is originally proposed for machine translation. Previous work also finds model ensemble, though not as practical due to high computational cost, a strong baseline for reducing model update regression. We also include it for comparison purpose.

We further propose a novel and generic method named Backward-Congruent Re-ranking (BCR). BCR takes into account the variety of structured prediction output. That is, a new model can produce a set of different predictions achieving similar accuracies. The key is to pick the one with the highest backward compatibility. In a nutshell, BCR uses the old model as a re-ranker to select a top structure from a set of candidates predicted by the new model. We further propose dropout-$p$ sampling, a simple and generic sampling method, to improve the diversity and quality of candidates. Despite the simplicity of BCR, it is a generic and surprisingly effective approach for mitigating model update regression, substantially outperforming knowledge distillation and ensemble methods across all studied model update settings. More surprisingly, we find that BCR can also improve the accuracy of the new model.

## 2  Related Work

***Forgetting* in Machine Learning**  Model update regression is related to a spectrum of research related to *forgetting* in machine learning, including continual learning (Chen and Liu, 2018; Parisi et al., 2019; Biesialska et al., 2020), incremental learning (Polikar et al., 2001; Gepperth and Hammer, 2016; Masana et al., 2020), which are concerned with the learning of new tasks or streaming examples and the forgetting of previously learned ones. The model update regression problem differs in that models are tested on the same task and have full access to previous data.

**Prior Research in Model Update Regression** The model update regression problem was first studied in (Shen et al., 2020) on learning backward-compatible visual embeddings for image retrieval. (Yan et al., 2021) suggested mitigating regression in image classification with positive-congruent training, a specialized variant of knowledge distillation. (Xie et al., 2021) extended the study of backward compatibility to NLP classification and explored several knowledge distillation variants for regression reduction. (Träuble et al., 2021) introduced a probabilistic approach to switch between the legacy and updated systems, bringing down regression in image classification. Our work investigates the model update regression in NLP structured prediction tasks, which can be particularly challenging due to the enormous output space and the variety of different factorizations.

**Re-ranking in Structure Prediction** The re-ranking technique has a long history in NLP for improving not only accuracy (Shen et al., 2004; Collins and Koo, 2005; Salazar et al., 2019; Yee et al., 2019), but also output diversity (Li et al., 2016), fluency (Kriz et al., 2019; Yin and Neubig, 2019), fairness (Geyik et al., 2019), and factual consistency (Falke et al., 2019). Different from prior work, we show that re-ranking can be adopted for improving backward compatibility in model updates.

## 3 Preliminaries

We first lay out the basic concepts for the quantitative measurement of model update regression in structured prediction. Then we introduce two representative structured prediction tasks in NLP: dependency parsing and conversational semantic parsing, and set up the evaluation benchmarks.

### 3.1 Prediction Flips in Structured Prediction

Model update regression happens after a *model update*, where an *old model* is replaced by a *new model*. For typical NLP structured prediction tasks (Smith, 2011), the input is a piece of natural language text and the output is a graph $G = (V, E)$ with multiple nodes $V$ and edges $E$. Unlike classification problems where the output is a label in a pre-defined label set, the output graph $G$ is decomposable and we may be interested in the correctness of some meaningful sub-graph $G' = (V' \subseteq V, E' \subseteq E)$ even though the graph is not completely correct. We call the graphs (or sub-graphs) that are correctly predicted by both the new and the old models as positive-congruent predictions. On the other hand, *negative flips* are the cases where the graphs (or sub-graphs) are correctly predicted by the old model but incorrectly predicted by the new one. There are also *positive flips*, where the new model corrects the mistakes that the old one makes, which are desirable accuracy improvements.

**Negative Flip Rate (NFR)** To measure the severity of model update regression, we count negative flips and compute their fraction of the total number of graphs (or sub-graphs). We term the fraction as *negative flip rate* (NFR) (Yan et al., 2021).

**Negative Flip Impact (NFI)** Note that NFR is capped by the overall error rate (ER) of the new model, comparison is challenging across cases with different ERs. For this reason, we introduce the *negative flip impact* (NFI): $\text{NFI} = \text{NFR}/\text{ER}_{\text{new}}$. Intuitively, the NFI computes the proportion of errors caused by negative flips. In other words, the proportion of errors that could have been avoided without the model update.

### 3.2 Dependency Parsing

The task of dependency parsing (Kübler et al., 2009) seeks to find the syntactic head word for each word in a sentence and their syntactic relation. The overall output has a tree structure where nodes are words in the sentence and edges are their relations. The importance of dependency parsing is widely recognized in the NLP community, with it benefiting a wide range of NLP applications (Qi et al., 2020).

We use the English EWT treebank from the Universal Dependency (UD2.2) Treebanks.[2] We adopt the standard training/dev/test splits and use the universal POS tags (Petrov et al., 2012) provided in the treebank. The parsing performance is measured by word-level accuracies: unlabeled/labeled attachment score (UAS/LAS), and tree-level accuracies: unlabeled/labeled complete match (UCM/LCM).

---

[2]http://universaldependencies.org/

We compute NFR and NFI under each accuracy metric. We find that the results of labeled metrics are always consistent with unlabeled metrics. For clarity, we only show the results of unlabeled metrics, the results of labeled metrics are presented in Appendix A. Following conventions (Dozat and Manning, 2017; Ma et al., 2018), we report results excluding punctuations.

### 3.3 Conversational Semantic Parsing

Conversational semantic parsing is an important component for intelligent dialog systems such as Amazon Alexa, Apple Siri, and Google Assistant, which transform user utterances into semantic frames comprised of intents and slots.

We use the TOP dataset (Gupta et al., 2018) for our experiments, which uses hierarchical intent-slot annotations for utterances in navigation and event domains.[3] The output is a constituency-based parse tree, where leaf nodes are words and the interior nodes are labeled with slots and intents. We use the dataset version with the noisy `IN:UNSUPPORTED_*` intents excluded (Einolghozati et al., 2019). The parsing performance is conventionally measured by tree-level Exact Match accuracy (EM). We also report span-level EM accuracy (span EM): We represent the semantic annotations on a sentence as a set of labeled spans $(i, j, L)$ where $i, j$ mark the start and end position of a span and $L$ is the corresponding label.[4] For example, the gold tree in Figure 1 can be viewed as four labeled spans including (0, 4, `IN:GET_DIRECTION`), (2, 4, `SL:DESTINATION;IN:FIND_EVENT`), (2, 2, `SL:ORGANIZER`), and (4, 4, `SL:CATEGORY`). We calculate the label accuracy on such spans. For example, the new model's output correctly predicted three of them, thus the span-level EM accuracy is $3/4 = 75\%$. We calculate NFR and NFI under both metrics.

## 4 Measuring Model Update Regression

We carry out a series of experiments to examine the severity of model update regression under various model update scenarios. In this section, we report the empirical results and conclude the findings.

### 4.1 Model Update Settings

In total, we consider four different causes for model updates.

- Change the training configurations (e.g., random seed, learning rate, etc). Despite this setting being less likely in practice, we consider it as a synthetic evaluation to understand the severity of model update regression.
- Scale up model sizes (e.g., increasing the depth or width of neural network). It is common that using a larger pre-trained language model brings in a larger performance gain.
- Train on more data. Practitioners frequently collect more and more labeled data for training.
- Take another factorization choice. It happens when new algorithms or models are invented and adopted.

### 4.2 Dependency Parsing

**Setup** For dependency parsing, we consider two popular models, namely the biaffine parser (Dozat and Manning, 2017) (*deepbiaf*) and the stack-pointer parser (Ma et al., 2018) (*stackptr*). The biaffine parser is a first-order graph-based parser, where the global prediction is decomposed to arc/edge predictions between any two words. The stack-pointer parser is a transition-based auto-regressive parser, where the global prediction is decomposed into a series of building action predictions. These two represent two distinct factorizations of the dependency parsing problem. Table 1 (top) presents their accuracies (mean and standard deviation). We can see that in terms of word-level accuracy, the performances of *deepbiaf* and *stackptr* are very close. However, in terms of complete match, *stackptr* is slightly better than *deepbiaf*, likely due to its auto-regressive nature.

We study model update regression in three different model update settings ( *deepbiaf*⇒*deepbiaf*, *stackptr*⇒*stackptr*, and *deepbiaf*⇒*stackptr*). In the first two settings, the old and new models have

---

[3]`http://fb.me/semanticparsingdialog`
[4]The label can be a chain of slot and/or intent labels.

| Model | UCM ↑ | UAS ↑ |
|---|---|---|
| *deepbiaf* | 63.88±0.56 | 91.76±0.10 |
| *stackptr* | 65.83±0.44 | 91.81±0.09 |

| Model Update | NFR ↓ | NFI ↓ | NFR ↓ | NFI ↓ |
|---|---|---|---|---|
| *deepbiaf*⇒*deepbiaf* | 3.67 | 10.17 | 1.55 | 18.80 |
| *stackptr*⇒*stackptr* | 3.44 | 10.07 | 1.64 | 19.99 |
| *deepbiaf*⇒*stackptr* | 3.84 | 11.24 | 2.14 | 25.57 |

Table 1: Accuracy (top) and model update regression (bottom) results on dependency parsing. ↑: higher is better and ↓: lower is better.

| Model | EM ↑ | span EM ↑ |
|---|---|---|
| *s2s-base*-part | 85.04±0.22 | 89.83±0.15 |
| *s2s-large*-part | 86.86±0.24 | 90.94±0.18 |
| *s2s-base* | 85.67±0.10 | 90.34±0.10 |
| *s2s-large* | 86.97±0.11 | 91.11±0.12 |

| Model Update | NFR ↓ | NFI ↓ | NFR ↓ | NFI ↓ |
|---|---|---|---|---|
| *s2s-base*-part⇒*s2s-base* | 2.98 | 21.12 | 2.25 | 23.68 |
| *s2s-large*-part⇒*s2s-large* | 2.61 | 20.03 | 1.99 | 22.37 |
| *s2s-base*⇒*s2s-large* | 3.32 | 25.45 | 2.61 | 29.39 |

Table 2: Accuracy (top) and model update regression (bottom) results on conversational semantic parsing. -part indicates the models are trained with only 4/5 of the available training data.

identical model architecture and training data. The only difference is that they are trained with different random seeds. We present these controlled experiments to demonstrate that even small changes such as altering random seeds can introduce significant regression.

**Results**   Table 1 (bottom) shows that model update regression is severe across all model update settings, including *deepbiaf*⇒*deepbiaf* and *stackptr*⇒*stackptr*. This implies that it could be fundamentally difficult to reduce model upgrade regression as initialization and optimization processes naturally introduce prediction inconsistencies. On the other hand, model update regression in heterogeneous model update (*deepbiaf*⇒*stackptr*) is even more severe. Particularly, negative flips account for up to 25% of the errors made by new models (*deepbiaf*⇒*stackptr* according to UAS). Notably, despite that *stackptr* improves over *deepbiaf* by 1.95% on UCM, both NFR and NFI in *deepbiaf*⇒*stackptr* is higher than those in homogeneous model updates. This demonstrates that reducing the error rate is not sufficient to reduce regression.

### 4.3   Conversational Semantic Parsing

**Setup**   For conversational semantic parsing, we use the *seq2seq* parser (Rongali et al., 2020) in our experiments. It formulates the semantic parsing task as *seq2seq* generation of the linearized parse tree given the source text. The global prediction is thus decomposed into a sequence of next-word predictions.

The *seq2seq* parsers can be initialized with different pre-trained language models. We experiment with two model variants: *seq2seq* parser with (1) `roberta-base` (Liu et al., 2019) (*s2s-base*) and (2) with `roberta-large` (*s2s-large*). In order to simulate data-driven model updates, we also train parsers with only 4/5 of the available training data, denoted by *s2s-base*-part and *s2s-large*-part. Table 2 (top) shows that *s2s-large* performs better than *s2s-base* and training on more data leads to better performance, in terms of both EM and span EM. We test three kinds of model updates: *s2s-base*-part⇒*s2s-base*, *s2s-large*-part⇒*s2s-large*, and *s2s-base*⇒*s2s-large*.

**Results**   As shown in Table 2 (bottom), model update regression is prevalent across different model update settings. The most severe model update regression comes from different pre-trained language models (*s2s-base*⇒*s2s-large*), accounts for 25.45% (EM) and 29.39% (span EM) errors made by the new model, while the absolute performance gains are merely 1.3% EM points and 0.77% span EM points.

### 4.4   Discussions

The key findings observed across all studied tasks and model updates are the following: (1) Model update regression is prevalent on two typical structured prediction tasks and various model update settings. (2) Even small changes in training configuration such as changing the random seed can introduce significant regression (a minimum of 1.55% NFR is observed). (3) Heterogeneous model updates lead to severer regression compared to homogeneous model updates. (4) Updating pre-trained language models leads to higher regressions than those caused by training on more data. As much as 3 out of 10 test errors are regression errors (NFI≈30%). (5) Improving accuracy alone does not necessarily reduce regression (in fact, NFR can be larger than accuracy gain).

# 5 Reducing Model Update Regression

First, we describe the adaptation of the methods explored in reducing model update regression in classification problems (Shen et al., 2020; Yan et al., 2021; Xie et al., 2021). Then, we introduce a novel method, backward-congruent re-ranking that is better suited to structured prediction tasks.

## 5.1 Model Ensemble

In classification problems, (Yan et al., 2021; Xie et al., 2021) found that model ensemble can reduce model update regression without any information about the old model. This can be attributed to the reduction of variance by ensembling: Every single model may capture the training data from a distinct aspect. The ensemble aggregates different aspects and has a shorter average distance to other single models. Concretely, we train different models with different random seeds and average the local prediction scores from all the models during decoding. Nevertheless, ensembles may not be practical due to the high cost of training and inference.

We note that parameter averaging (i.e., averaging the parameters of different models and using the resulting model for prediction) is an alternative to prediction ensembling. However, our preliminary experiments show that this method gives worse performance than prediction ensembling.

## 5.2 Knowledge Distillation

Knowledge distillation (KD) is a technique originally proposed for model compression (Buciluǎ et al., 2006; Ba and Caruana, 2014; Hinton et al., 2015), where a (smaller) *student* model is trained to mimic a (larger) *teacher* model. By treating the old model as the teacher and the new model as the student, KD has been proved a promising approach for reducing model update regression in both vision and language classification tasks (Yan et al., 2021; Xie et al., 2021). The instantiations are usually based on a loss function computing the cross-entropy between the output distributions predicted by the teacher and the student. However, for structured prediction problems, the output space is exponential in size, making the exact output distribution computationally intractable. Although some efficient methods for approximating the cross-entropy have been proposed (Zmigrod et al., 2021a; Wang et al., 2021), they assume the models follow certain factorizations and thus have limited application scenarios (e.g., the techniques proposed in (Zmigrod et al., 2021a) can only be used for edge-factored parsers). One may think of performing KD on local predictions or hidden representations instead. However, such distillation can also be infeasible between models with different factorizations or structures (e.g., *deepbiaf* ⇒ *stackptr* in dependency parsing).

To tackle the above problem, we borrow the idea from sequence-level KD (Kim and Rush, 2016), which approximates the teacher distribution with its mode. Specifically, sequence-level KD suggests to (1) run the teacher model over the training set to create pseudo training data, (2) then train the student model on this new dataset. The idea is well-suited to the problem of model update regression because it is model-agnostic; it does not assume any specific factorizations for the old and new models.

## 5.3 Backward-Congruent Re-ranking

Unlike model ensemble, knowledge distillation attempts to explicitly align the behaviors of the new model to the old model during *training*. Alternatively, we propose Backward-Congruent Re-ranking (BCR), which does not impose any constraint on the training of the new model and only takes effect during *inference*.

Re-ranking is a popular approach in structured prediction to combine the strengths of two different models (Collins and Koo, 2005; Socher et al., 2013; Le and Zuidema, 2014; Do and Rehbein, 2020). It suggests to use one model as the *candidate generator* for creating a candidate pool and use the other model as the *re-ranker* for picking the best candidate. While previous work has been focused on developing powerful re-rankers for overall performance improvement, the purpose of BCR is to reduce model update regression. BCR treats the new model as the candidate generator and the old model as the re-ranker. Our motivations are two-fold. First, structured prediction models can produce many possible predictions. Second, different predictions may achieve similar error rates but differ by

the mistakes they made. Among them, the most likely one according to the old model should have the least prediction flips. These make re-ranking particularly useful for structured prediction.

Formally, we have the old model and the new model parameterized by $\phi_{new}$ and $\phi_{old}$ respectively. For a given input $x$, the new model first generates a set of candidates $\text{GEN}_{\phi_{new}}(x)$. Then we choose the prediction $y^*$ with the highest score computed by the old model.

$$y^* = \underset{y \in \text{GEN}_{\phi_{new}}(x)}{\arg\max} \ p_{\phi_{old}}(y|x),$$

where $p_{\phi_{old}}(y|x)$ is the old model's generation probability (score) of $y$ given the input $x$ and GEN can be implemented by various decoding methods, including maximization-based search such as beam search and $k$-best MST algorithm (Zmigrod et al., 2020, 2021b), and stochastic sampling such as top-$k$ sampling (Fan et al., 2018; Radford et al., 2019) and top-$p$ sampling (Holtzman et al., 2019).

**Dropout-$p$ Sampling**    Maximization-based search attempts to find the top outputs that received the highest scores from the model. This inherently leads to a set of similar candidates and re-ranking may suffer from the lack of diversity. On the other hand, stochastic sampling introduces more variances among candidates by randomized choices during decoding. However, traditional stochastic sampling may suffer from the low quality of sampled outputs due to the deterministic nature of structured prediction. Furthermore, it is unclear how to adapt popular sampling methods, such as top-$k$ and top-$p$ sampling, to solutions that are not formalized as sequence generation (*e.g.*, the biaffine parser).

In this work, we explore a special sampling method, dropout-$p$ sampling, that has attracted little attention in the literature. Dropout (Srivastava et al., 2014) is a regularization technique used in almost all modern neural network-based models. The key idea is to randomly drop some neurons from the neural network during training. Normally, dropout is turned off during inference. However, in dropout-$p$ sampling, we keep using dropout with dropout rate $p$. Compared to traditional sampling methods, dropout-$p$ sampling has the following advantages: (1) It directly changes the scoring function and keeps any default decoding algorithm unchanged; (2) It can be regarded as conducting global sampling instead of a series of local sampling at each decoding step, potentially improving the formality of output structure; (3) It also has a broader applicable scope that is not limited to sequence generation models.

**Discussion**    Following previous work (Shen et al., 2004; Yan et al., 2021; Xie et al., 2021), our discussion has been focused on handling one model update. However, BCR can be extended to handle multiple turns of model updates. To do so, one can keep the most recent $k$ models and use a weighted combination of their scores as the re-ranking metric. In practice, $k$ can be set to trade-off between performance and runtime cost.

One downside of BCR is that we must maintain and deploy both the old and new models. Because the re-ranking step has less time complexity compared to the decoding algorithms and the computation is fully parallelizable, this does not create much additional inference latency. However, the increase in memory footprint does entail an increase in the inference hosting cost. One remedy could be to use knowledge distillation to distill the old model(s) into a smaller one, which we leave for future work.

## 6   Experiments

### 6.1   Setup

For the ensembles of biaffine parsers, we first average the edge scores from each model, then run MST on the average scores. For stack-pointer and *seq2seq* parsers, we average the local prediction scores from each model at each decoding step. We combine 5 models in model ensemble. The impact of ensemble size is discussed in Appendix B. For knowledge distillation, as mentioned in Section 5.2, we simply replace the ground-truth output in the original training set with the old model's predictions when training the new model.[5] For BCR, various decoding methods are explored for candidate generation. Specifically, we use $k$-best spanning trees algorithm (Zmigrod et al., 2020, 2021b) with biaffine parsers, and beam search with stack-pointer parsers and *seq2seq* parsers. We also explore

---

[5]We also tried the combination of pseudo and original training data but found little difference in performance and regression.

| | deepbiaf⇒deepbiaf | | | | | | stackptr⇒stackptr | | | | | | deepbiaf⇒stackptr | | | | | |
|---|---|---|---|---|---|---|---|---|---|---|---|---|---|---|---|---|---|---|
| | UCM | | | UAS | | | UCM | | | UAS | | | UCM | | | UAS | | |
| | ACC | NFR | NFI | ACC | NFR | NFI | ACC | NFR | NFI | ACC | NFR | NFI | ACC | NFR | NFI | ACC | NFR | NFI |
| Old | 63.88 | - | - | 91.76 | - | - | 65.83 | - | - | 91.81 | - | - | 63.88 | - | - | 91.76 | - | - |
| ↪Untreated | 63.97 | 3.69 | 10.25 | 91.64 | 1.66 | 19.85 | 66.03 | 3.43 | 10.10 | 91.73 | 1.67 | 20.20 | 66.03 | 3.73 | 10.98 | 91.73 | 2.10 | 25.37 |
| ↪Distillation | 64.00 | 3.82 | 10.62 | 91.67 | 1.62 | 19.45 | 65.66 | 3.57 | 10.40 | 91.70 | 1.70 | 20.49 | 66.11 | 3.62 | 10.68 | 91.78 | 2.03 | 24.71 |
| ↪Ensemble | **64.81** | 2.51 | 7.14 | **92.10** | 1.02 | 12.97 | **67.21** | 2.21 | 6.74 | **92.21** | 1.11 | 14.30 | **67.21** | 2.83 | 8.62 | **92.21** | 1.62 | 20.75 |
| ↪BCR | 64.36 | **1.12** | **3.14** | 91.78 | **0.84** | **10.21** | 66.45 | **1.05** | **3.12** | 91.88 | **0.84** | **10.30** | 66.76 | **1.20** | **3.60** | 92.01 | **1.11** | **13.87** |

Table 3: Comparison of different regression reduction methods on dependency parsing (NFR ↓ NFI ↓ ACC ↑) using unlabeled metrics (results of labeled metrics are shown in Appendix A). Old indicates the old model's performance before model update. Untreated denotes the results of new models without any treatment. BCR denotes the results with backward-congruent re-ranking.

sampling-based decoding methods such as top-$k$ sampling ($k \in \{5, 10, 50, 100\}$) (Fan et al., 2018; Radford et al., 2019), top-$p$ sampling ($p \in \{0.95, 0.90, 0.85, 0.80\}$) (Holtzman et al., 2019), and dropout-$p$ sampling ($p \in \{0.1, 0.2, 0.3, 0.4\}$). The number of candidates in BCR is set to 10. Our experiments show that the best results are always achieved using dropout-$p$ sampling. For brevity, we only report the best results and defer the detailed analysis of the candidate generation methods to Section 6.4. We include the model update regression results without any treatment in Section 4 (denoted as Untreated) for reference (i.e., the new models are trained normally with a different set of random seeds).

For dependency parsing, we adapt the implementation of the biaffine parser Dozat and Manning (2017) and the stack-pointer parser Ma et al. (2018) in NeuroNLP2[6]. We use the default configurations suggested in (Ma et al., 2018) for training and testing models. For conversational semantic parsing, we re-implement the *seq2seq* parser Rongali et al. (2020) using Fairseq Ott et al. (2019) and the pre-trained language models Liu et al. (2019) are downloaded from HuggingFace[7].

## 6.2 Results on Dependency Parsing

Table 3 shows the results of different methods for reducing model update regression on the dependency parsing task. We can see that model ensemble brings down negative flips in all model update settings compared with the untreated baseline (on average $28\%$ relative reduction in NFI). On the other hand, knowledge distillation shows little effect on reducing model update regression. We hypothesize that it is due to the almost perfect word-level accuracy (UAS) of the old model when running on the training set. As a result, there is little difference between the original training set and the synthetic training set used for distillation, leading to performance close to the untreated baseline.

For BCR, it substantially reduces NFR and NFI across all model update settings (an average of $58\%$ relative reduction in NFI), greatly outperforming model ensemble and knowledge distillation. This is very encouraging given that model ensemble is often the most effective approach to reduce regression in classification problems (Yan et al., 2021; Xie et al., 2021) or even considered as a paragon (Yan et al., 2021). Interestingly, BCR delivers more pronounced gains over model ensemble in the heterogeneous model update setting (*deepbiaf⇒stackptr*). In particular, BCR reduces NFI by $67\%$ while model ensemble only obtains $21\%$ relative reduction, according to UCM. The reason might be that model ensemble can only reduce the variance in models of the same kind, while the intrinsic gap between different kinds of models cannot be reduced. In contrast, BCR selects the most backward-compatible prediction regardless of the source of candidates. Another interesting observation is that BCR can also improve the overall accuracies though the improvements slightly lag behind model ensemble (e.g., $+0.19$ vs. $+0.37$ UAS points on average). We regard it as a side benefit since our primary goal here is to reduce model update regression rather than to boost accuracy.

## 6.3 Results on Conversational Semantic Parsing

Table 4 shows the results of different methods for reducing model update regression on the conversational semantic parsing task. The empirical results generally support the main findings on the

---

[6] https://github.com/XuezheMax/NeuroNLP2
[7] https://huggingface.co

| | s2s-base-part⇒s2s-base | | | | | | s2s-large-part⇒s2s-large | | | | | | s2s-base⇒s2s-large | | | | | |
|---|---|---|---|---|---|---|---|---|---|---|---|---|---|---|---|---|---|---|
| | EM | | | span EM | | | EM | | | span EM | | | EM | | | span EM | | |
| | ACC | NFR | NFI | ACC | NFR | NFI | ACC | NFR | NFI | ACC | NFR | NFI | ACC | NFR | NFI | ACC | NFR | NFI |
| Old | 85.04 | - | - | 89.83 | - | - | 86.86 | - | - | 90.94 | - | - | 85.67 | - | - | 90.34 | - | - |
| ↪Untreated | 85.88 | 2.98 | 21.12 | 90.50 | 2.25 | 23.68 | 86.98 | 2.61 | 20.03 | 91.10 | 1.99 | 22.37 | 87.04 | 3.16 | 24.40 | 90.96 | 2.68 | 29.64 |
| ↪Distillation | 85.54 | 2.61 | 18.06 | 90.23 | 2.07 | 21.21 | 86.64 | 2.04 | 15.25 | 90.88 | 1.63 | 17.90 | 87.69 | 1.97 | 15.97 | 91.53 | 1.79 | 21.16 |
| ↪Ensemble | **86.90** | 1.73 | 13.24 | **91.35** | 1.28 | 14.80 | **87.65** | 1.17 | 9.47 | **91.62** | 1.00 | 11.78 | **87.65** | 2.70 | 21.88 | **91.62** | 2.15 | 25.60 |
| ↪BCR | 86.16 | **0.75** | **5.39** | 90.69 | **0.69** | **7.42** | 87.41 | **0.54** | **4.30** | 91.24 | **0.67** | **7.61** | 87.49 | **0.69** | **5.51** | 91.53 | **0.74** | **8.78** |

Table 4: Comparison of different methods on conversational semantic parsing (NFR ↓ NFI ↓ ACC ↑).

| Method | EM | | | span EM | | |
|---|---|---|---|---|---|---|
| | ACC | NFR | NFI | ACC | NFR | NFI |
| Untreated | 87.04 | 3.16 | 24.40 | 90.96 | 2.68 | 29.64 |
| Beam, $b$=10 | 87.61 | 1.63 | 13.20 | 91.52 | 1.49 | 17.57 |
| top-$k$=5 | 87.71 | 2.16 | 17.55 | 91.62 | 1.81 | 21.63 |
| top-$p$=0.95 | **87.73** | 2.18 | 17.77 | **91.66** | 1.81 | 21.65 |
| dropout-$p$=0.3 | 87.49 | **0.69** | **5.51** | 91.53 | **0.74** | **8.78** |
| dropout-$p$=0.1 | 88.01 | 1.26 | 10.54 | **91.98** | 1.06 | 13.23 |
| dropout-$p$=0.2 | 87.87 | 0.88 | 7.27 | 91.88 | 0.81 | 10.03 |
| dropout-$p$=0.3 | 87.49 | 0.69 | 5.51 | 91.53 | 0.74 | 8.78 |
| dropout-$p$=0.4 | 87.26 | 0.72 | 5.63 | 91.31 | 0.81 | 9.36 |

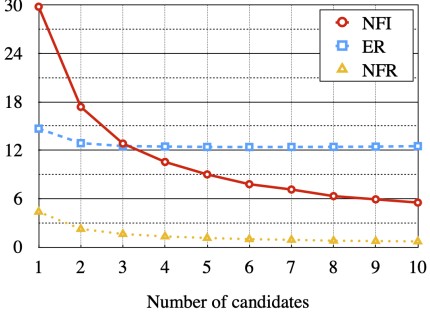

Table 5: Comparison of different decoding methods with selected parameters of each method (*s2s-base*⇒*s2s-large*).

Figure 2: Effect of increasing number of candidates to NFI (↓), NFR (↓), and error rate (ER ↓) (*s2s-base*⇒*s2s-large*).

dependency parsing task: (1) Knowledge distillation has little effect on reducing NFR and NFI (except for the *s2s-base*⇒*s2s-large* setting). (2) Model ensemble brings moderate reduction (28% relative NFI reduction on average) across all model update settings. (3) BCR achieves substantial reduction (73% relative NFI reduction on average), greatly outperforming model ensemble and knowledge distillation. BCR not only reduces model update regression but also improves accuracies as compared to the untreated baseline. This is particularly intriguing since we are using a weaker model (the old model) is used to re-rank the candidates generated by a stronger model (the new model). We speculate that the reason is the complementary effect of different models in differentiating distractors.

## 6.4 Analysis of BCR

Next, we study the two key components of BCR, namely candidate generation and the re-ranking metric. We analyze the results of different decoding methods described in Section 5.3 and empirically reveal the performance upper-bounds of BCR.

**Candidates for Re-ranking** We present the results of BCR with different decoding methods in Table 5. For clarity, we only report the best result obtained using each decoding method together with the optimal hyper-parameter setting (except for dropout-$p$ sampling). We can see that dropout-$p$ sampling obtains the largest reduction in both NFR and NFI, substantially outperforming beam search and traditional truncated sampling (the gaps in NFI are larger than 10 points). top-$p$ and top-$k$ sampling are the least effective approaches. We hypothesize that traditional truncated sampling is not suitable for our structured prediction tasks. Unlike open-ended generation tasks, there are only a few valid local predictions at each decoding step. Therefore, the candidates generated by truncated sampling may lack diversity. For dropout-$p$ sampling, as $p$ increases from 0.1 to 0.3, the improvements in accuracies shrink while the reductions in NFR and NFI increase. This is expected since a larger $p$ leads to lower quality of individual candidates but a more diverse candidate pool.

We plot the curves of NFR, NFI, and error rate (ER) with the number of candidates in Figure 2. As seen, all three metrics decrease as the number of candidates increases. However, they converge at different points. NFI and NFR drop significantly even after the reduction in ER reaches its limit.

**Oracle Re-ranker** To shed light on the effectiveness of our choice of the re-ranking metric (i.e., the old model's prediction score $p_{\phi_{old}}$), we compare it to two kinds of oracle re-rankers: one using span-level EM (w/ ACC) and one using span-level NFR (w/ NFR). The ACC and NFR re-rankers

represent the upper bounds of ACC and NFR that re-ranking methods can achieve. The results are presented in Table 6. It can be observed that the NFI and NFR of using $p_{\phi_{\text{old}}}$ are very close to the upper bounds for reducing model update regression (w/ NFR). Another notable observation is that the NFIs of using $p_{\phi_{\text{old}}}$ and NFR re-ranking are much lower than the NFI of ACC re-ranking. This demonstrates that improving accuracy is not equivalent to reducing model update regression.

**Inference Speed**  For dropout-$p$ sampling, the overall computation overhead of the decoding step grows linearly with the number of candidates. Nevertheless, different runs of sampling can be done in parallel. With the same inference hardware (one Nvidia V100 GPU) and the same batch size of 32, the *decoding* and *re-ranking* speeds of *deepbiaf* are 171 and 244 sentences per second, and 64 and 221 sentences per second for *stackptr*. For *seq2seq* parsers, we observe that the speed of re-ranking is about 5x of the decoding speed. In practice, the re-ranking step can be made even faster as it generally allows for larger batch sizes.

| Re-ranker | EM | | | span EM | | |
|---|---|---|---|---|---|---|
| | ACC | NFR | NFI | ACC | NFR | NFI |
| Untreated | 87.04 | 3.16 | 24.40 | 90.96 | 2.68 | 29.64 |
| w/ $p_{\phi_{\text{old}}}$ | 87.49 | 0.69 | **5.51** | 91.53 | 0.74 | 8.78 |
| w/ NFR | 89.50 | **0.65** | 6.21 | 92.90 | **0.58** | **8.13** |
| w/ ACC | **92.68** | **0.65** | 8.90 | **95.04** | 0.59 | 11.82 |

Table 6: Comparison of BCR ($p_{\phi_{\text{old}}}$) to oracle re-rankers (NFR, ACC) (*s2s-base⇒s2s-large*).

## 7  Conclusions

This paper presented the first study on the model update regression issue in NLP structured prediction tasks. Experiments on two representative structured prediction tasks showed that model update regression is severe and widespread across different model update settings. To reduce model update regression, we explored knowledge distillation, model ensemble, and backward-congruent re-ranking. Backward-congruent re-ranking consistently achieves much more significant regression reduction than distillation and ensemble methods. In the future, model update regression should be examined in other structured prediction tasks such as text generation in NLP and image segmentation in computer vision, and scenarios of multiple rounds of model updates.

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
