# A  Results on Dependency Parsing (Labeled Metrics)

The results with labeled metrics (LCM and LAS) are presented in Table 7 and Table 8. We can see that the results exhibit patterns that are consistent to the results with unlabeled metrics. The main findings in Section 6.2 are general across labeled and unlabeled measurement: model ensemble can reduce model update regression to some extent, knowledge distillation has little effect, and backward-congruent re-ranking brings the most substantial reduction. Specifically, both model ensemble and backward congruent re-ranking reduce model update regression across all model update settings. The reduction that comes from backward-congruent re-ranking is consistently larger than model ensemble. Backward-congruent re-ranking also improves accuracies as compared to the untreated baseline, though the improvements are slightly lower than model ensemble.

| Model | LCM ↑ | LAS ↑ |
|---|---|---|
| *deepbiaf-part* | 55.87±0.44 | 88.87±0.13 |
| *stackptr-part* | 58.11±0.76 | 88.88±0.23 |
| *deepbiaf* | 57.43±0.52 | 89.66±0.11 |
| *stackptr* | 59.08±0.65 | 89.62±0.13 |

| Model Update | NFR ↓ | NFI ↓ | NFR ↓ | NFI ↓ |
|---|---|---|---|---|
| *deepbiaf⇒deepbiaf* | 3.28 | 7.71 | 1.72 | 16.65 |
| *stackptr⇒stackptr* | 3.37 | 8.22 | 1.84 | 17.73 |
| *deepbiaf⇒stackptr* | 3.69 | 9.02 | 2.35 | 22.63 |

Table 7: Accuracy (top) and model update regression (bottom) results on dependency parsing. ↑: higher is better and ↓: lower is better.

| | *deepbiaf⇒deepbiaf* | | | | | | *stackptr⇒stackptr* | | | | | | *deepbiaf⇒stackptr* | | | | | |
|---|---|---|---|---|---|---|---|---|---|---|---|---|---|---|---|---|---|---|
| | LCM | | | LAS | | | LCM | | | LAS | | | LCM | | | LAS | | |
| | ACC | NFR | NFI | ACC | NFR | NFI | ACC | NFR | NFI | ACC | NFR | NFI | ACC | NFR | NFI | ACC | NFR | NFI |
| Old | 57.43 | - | - | 89.66 | - | - | 59.08 | - | - | 89.62 | - | - | 57.43 | - | - | 89.66 | - | - |
| ↪Untreated | 57.48 | 3.28 | 7.72 | 89.53 | 1.83 | 17.52 | 59.30 | 3.42 | 8.40 | 89.52 | 1.89 | 18.00 | 59.30 | 3.59 | 8.82 | 89.52 | 2.39 | 22.78 |
| ↪Distillation | 57.75 | 3.14 | 7.43 | 89.60 | 1.77 | 17.06 | 58.77 | 3.70 | 8.97 | 89.45 | 1.94 | 18.36 | 59.13 | 3.64 | 8.91 | 89.55 | 2.33 | 22.25 |
| ↪Ensemble | **58.55** | 2.08 | 5.03 | **90.12** | 1.10 | 11.17 | **60.64** | 2.13 | 5.41 | **90.18** | 1.21 | 12.36 | **60.64** | 2.62 | 6.65 | **90.18** | 1.77 | 18.04 |
| ↪BCR | 57.91 | **1.02** | **2.43** | 89.69 | **0.99** | **9.60** | 59.72 | **1.11** | **2.74** | 89.71 | **1.01** | **9.81** | 60.02 | **1.15** | **2.88** | 89.90 | **1.31** | **12.98** |

Table 8: Comparison of different regression reduction methods on dependency parsing (NFR ↓ NFI ↓ ACC ↑) using **labeled metrics** (LCM and LAS). Old indicates the old model's performance before model update. Untreated denotes the results of new models without any treatment. Distillation, Ensemble, and BCR denote the results with distillation, ensemble, and backward-congruent re-ranking respectively.

# B  Impact of Ensemble Size

We report the impact of ensemble size in Figure 3. As seen, there is no large performance boost with a larger ($> 5$) ensemble size. Ensembling 10 models still underperforms BCR, though computational cost grows as a multiplier of the number of models in an ensemble. We have similar observations across all model update settings.

# C  Oracle Re-rank (*deepbiaf⇒stackptr*)

Following the analysis in Section 6.4, we present additional results of oracle re-rankers for better understanding of the performance of BCR. Concretely, we report the results on the heterogeneous model update (i.e., *deepbiaf⇒stackptr*) in dependency parsing. In Table 9, we compare BCR to two kinds of oracle re-rankers: one using UAS (w/ ACC) and one using UAS NFR (w/ NFR). The ACC and NFR re-rankers represent the upper bounds of ACC and NFR that re-ranking methods can achieve. As seen, the NFI and NFR of using $p_{\phi_{\text{old}}}$ are close to the upper bounds for reducing model

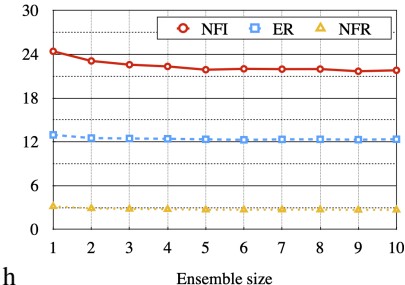

h

Figure 3: Comparison of different ensemble size (*s2s-base*⇒*s2s-large*).

update regression (w/ NFR). Using $p_{\phi_{\text{old}}}$ and NFR re-ranking can obtain lower NFI than using ACC re-ranking. The results are consistent to the observations in Section 6.4.

| Re-ranker | UCM | | | UAS | | |
|---|---|---|---|---|---|---|
| | ACC | NFR | NFI | ACC | NFR | NFI |
| Untreated | 66.03 | 3.73 | 10.98 | 91.73 | 2.10 | 25.37 |
| w/ $p_{\phi_{\text{old}}}$ | 66.76 | 1.20 | 3.60 | 92.01 | 1.11 | 13.87 |
| w/ NFR | 70.85 | **0.98** | **3.35** | 93.53 | **0.65** | **10.11** |
| w/ ACC | **75.09** | **0.98** | 3.92 | **94.90** | 0.76 | 15.00 |

Table 9: Comparison of BCR ($p_{\phi_{\text{old}}}$) to oracle re-rankers (NFR, ACC) (*deepbiaf*⇒*stackptr*).