# OpenReview forum: "Measuring and Reducing Model Update Regression in Structured Prediction for NLP"
_NeurIPS.cc/2022/Conference — NeurIPS 2022 Accept_

### Official Review · Reviewer_EpNN · 2022-07-02

**Rating:** 6
**Confidence:** 4
**Soundness:** 3 good
**Presentation:** 4 excellent
**Contribution:** 2 fair

**Summary:**

This paper tackles the problem of model update regression: when a model with better aggregate performance is deployed, it may make mistakes that a previous model didn't make.  How can we minimize these "regressions"?  Specifically, this paper is the first to look at this problem in a structured prediction setting. On both dependency parsing and on a conversational dialogue / semantic parsing task, the paper evaluates new models to see how many examples they regress on. The paper describes several techniques to mitigate regression, including a new one, backward-congruent reranking, that reranks the new model's output using the old model. This is combined with a form of dropout at test time to yield diverse samples. The paper shows that BCR achieves good task accuracy while minimizing the number of model regressions.

**Questions:**

How is RoBERTa used for the seq2seq semantic parser? Is this the encoder and a separate transformer decoder model is learned? Please clarify since this is not a standard use of the encoder-only RoBERTa model.

**Limitations:**

There is little discussion of limitations and societal impact, but I don't see this as a big problem for this paper.

**Strengths And Weaknesses:**

STRENGTHS

- The problem setting in this paper is novel. While a somewhat niche problem, I am convinced of the core argument that this could be useful (with some reservations; see below under "Weaknesses").

- The exploration of "new" models is quite thorough, and it's nice to see the contrasts between some different types of model updates, including changing the mode paradigm.

- The BCR technique is simple and I could see it being widely adopted for this task

- The dropout-p sampling technique is interesting and a novel way of getting diverse samples from these kinds of generation models.

WEAKNESSES

The main conceptual weakness I see in this paper is the notion of a regression as it ties to model confidence.

It seems to me that if you take a model achieving 90% accuracy and perturb its weights slightly, it may still get 90% accuracy but make different predictions.  Many of the flipped predictions will be those where the 2nd-best option had nearly the same probability as the 1-best option; that is, cases where the confidence was low / entropy of the label distribution was high. It's not clear to me in these cases that the first model is getting things "right" and the second is getting them "wrong", more that the model had high uncertainty and it broke one way (correct) for some examples and another way (incorrect) for other examples.

Considering that BCR's performance is always lower than that of the ensemble, it seems to me that what's happening here is that this is really an ensemble that weights more heavily towards the earlier model. We aren't ensembling the two models symmetrically, but by reranking outputs from the second using the first, we're really only making changes where the second model is significantly more confident and doesn't even return what the first model initially preferred, resulting in a smaller performance gain but also fewer changes. So basically this is a clever way of combining the models but mostly has the effect of preferring the first.

I also wonder about the NFR/NFI metrics as they relate to the motivation of this paper. The idea of a system changing and then breaking UX or downstream modules that were depending on certain things working is a problem that resonates with me. However, do NFI and NFR really correlate with this motivation?  Again, many of the differences seem like they could most heavily depend on examples that were basically half-working to begin with, which were probably not consistent features a user was very attached to.

As a result, while this paper explores a different point in the design space of how to ensemble these two models, I'm not sure what it presents is really groundbreaking in terms of results.

---

> ### Author Response · Authors · 2022-08-01
> **Response to Reviewer EpNN**
>
> Thanks for your thoughtful review.
>
> For the relationship between regression and model confidence. In a previous work on image classification[1, Section 5.4], they have an in-depth analysis. The major conclusion is that test samples that become negative flips have relatively high estimated uncertainty but the uncertainty measure does not clearly separate them from other samples. Regardless of the relationship, we do not see how it can be a weakness of our paper. The reason is that our ultimate goal is to improve backward-compatibility for end users. We want an user to continue to receive the “right” predictions that could be achieved by the old model.
>
> In addition to your valuable thoughts on the interpretation of BCR, we would like to mention that BCR also improves the accuracy compared to the old model. This is particularly intriguing when we are using a weaker model (the old model) re-rank the candidates generated by a stronger model (the new model). We speculate that the reason is the complementary effect of different models in differentiating distractors.
>
> In our understanding the reviewer is talking about two different topics. *Question1: Does NFI/NFR really correlate with “human-judgment” of the model working or not?* Our answer: UX and human perception of how well the model works is a separate metric to consider (more of a human-computer interaction question). NFR/NFI gives us a straightforward and easy way to measure the compatibility of the model without running expensive human case studies. Measuring negative flips is a common practice for model update regression research. However, we agree that It would be interesting to have a deeper analysis on user experience. *Question2: If the model is half-working to begin with, does the differences between models really matter?* Our answer: We agree that if the old model is half-working to begin with, the regression benchmarking doesn’t make sense. And rather the developer should prioritize making sure that the new update gets as high as possible accuracy to have a pleasant experience for the customers. That is why in this paper we only benchmarking model updates and regressions that already have high enough accuracy (80%+) to be used by the users.
>
> For your question on the seq2seq semantic parser, we initialize the encoder with RoBERTa and use a randomly-initialized decoder, following [2].
>
> [1] Positive-Congruent Training: Towards Regression-Free Model Updates. CVPR2021
>
> [2] Don’t parse, generate! A sequence to sequence architecture for task-oriented semantic parsing. WWW2020

---

### Official Review · Reviewer_XYXe · 2022-07-10

**Rating:** 4
**Confidence:** 4
**Soundness:** 3 good
**Presentation:** 4 excellent
**Contribution:** 3 good

**Summary:**

The paper studied a problem called model update regression in the field of structured prediction for NLP, which means a new model that achieves higher scores may perform worse than its baselines on some cases. The authors proposed an approach, BCR, to handle this issue. The basic idea is to use an old model to filter predicted candidates from the new model. A trick called dropout-p sampling is also applied to improve the diversity and quality of produced candidates. The experiments had been done on Dependency Parsing and Conversational Semantic Parsing, which were aimed to show the effectiveness of BCR.

**Questions:**

1) Can you put some case studies in the Experiment section to show where your gains are from?
2) Since more than one model is used in your approach, does this strategy cause some of your improvements are in fact overlapping with the ensemble approach?

**Limitations:**

Besides my concerns raised in 'Strengths And Weaknesses', I think the authors should do more experiments to empirically confirm the problem and describe it in a more acceptable way. I have seen that the authors took a lot of space to show the performance numbers of their models, which I believe should not be the focus of this paper.

**Strengths And Weaknesses:**

Pros:
1) the writing is good and clear;
2) the problem is well defined and the solution is straightforward.

Cons:
1) syntactic parsing is indeed very classical for computational linguistics but only stands for a tiny part of tasks in the field of structured prediction. Some other tasks like NER and POS Tagging are far more common and have much more applications in the NLP industry. Plus, a lot of space in the paper is used for introducing syntactic parsing, which causes a bar even for researchers from NLP and is not necessary for its future applications to common scenarios. Therefore, The authors should investigate more representative tasks in structured prediction such that the formulation is more clear and the work has a broad impact on the whole community.
2) knowledge distillation for structured prediction is a mature field in NLP. Please at least see this work, Structure-Level Knowledge Distillation For Multilingual Sequence Labeling, and check the recent literature. Hence, the authors shouldn't say like 'existing knowledge distillation-based solutions cannot be directly applied to structured prediction models' in the paper, and more solid experiments are expected for comparisons.
3) the problem mentioned in the paper is formulated in a quite intuitive manner, which might lead to many explanations that can't be proved false. For example, from my view, the problem may be a cause of randomness in the premise that neural networks can't fit the whole dataset, which may explain why the ensemble model works. Therefore, I believe a theoretical foundation is expected.

---

> ### Author Response · Authors · 2022-08-01
> **Response to Reviewer XYXe**
>
> Thank you for your review!
>
> ***[representative tasks in structured prediction]***
> Thank you for your thoughtful comments. First of all, we would like to clarify that our work studied both syntactic parsing and semantic parsing. Conversational semantic parsing is a very practical problem used in conversational assistants Alexa, Siri, and etc. The GOAL was to consider one fundamental (syntactic dependency parsing) and one practical task (conversational semantic parsing). The most important characteristic of our tasks is that the output is graph-structured (tree-structured). In contrast, the output of NER and POS tagging is word-level tags. In fact, NER and POS belong to a more fine-grained category of sequence labeling. Compared to sequence labeling, graph-structured prediction is much more complicated in terms of structural complexity (capturing structural dependencies among outputs is a MUST in a parsing task, while such dependencies are weaker in NER or POS). Graph-structured prediction is the standard formalization for syntactic parsing and semantic parsing, and plays a large part in the whole NLP community.  We are considering changing “structured prediction” to “graph-structured prediction” to make our contribution clearer.
>
> ***[knowledge distillation]***
> Thank you for pointing out the issue. We are sorry for the confusion likely caused by our writing. In fact, by “existing knowledge distillation methods”, we actually mean “global/vanilla knowledge distillation” (the classic definition of knowledge distillation from Hinton et al 2015). We will revise the writing. However, your suggested paper is on sequence labeling and particularly CRF-based models. Their methods are not directly applicable to our problems (graph-structured prediction) and our models (non-CRF models). One major challenge for structure prediction tasks is that the prediction can be factorized in many distinct ways. Such model-specific techniques cannot be used for model updates across different factorizations (e.g., heterogeneous model updates such as deepbiaf ⇒stackptr in dependency parsing). For more detailed discussion on why we strive to develop model-agnostic solutions, see [our discussion with Reviewer TTdd](https://openreview.net/forum?id=4cdxptfCCg&noteId=6GgxvORJXrF3). To make our claim clear to readers, we will soften the sentence "existing methods are no directly applicable to structure prediction tasks" and add more discussion in Sec. 5.2 about your suggested references. Thank you vey much for pointing out these missing references.
>
> ***[causes for regression]***
> Thank you for your insightful thoughts. Yes, randomness is part of the source for the regression problem. See Line 198-199, “Even small changes in training configuration such as changing random seed can introduce significant regression (a minimum of 1.55% NFR is observed)”. However, it is not the only source according to our other experiment setups (See Section 4.1). Changes in model architecture, factorization and data size can result in more severe regression. Also, the problem of randomness may not come from under-fitting, because we find the training accuracy for dependency parsers is close to 100%. Our hypothesis for model ensemble is given in Line 210-212,  “This can be attributed to the reduction of variance by ensembling: Every single model may capture the training data from a distinct aspect. The ensemble aggregates different aspects and has a shorter average distance to other single models.”
>
> ***[case studies]***
> We will add some case studies when more space is allowed.
>
> ***[model ensembling]***
> Thanks for the interesting question! Model ensembling doesn't make use of any information of the old model. Our proposed BCR explicitly uses the guidance from the old model. We speculate the overlap would not be significant. We are happy to run such experiments.

---

### Official Review · Reviewer_uHkD · 2022-07-10

**Rating:** 8
**Confidence:** 4
**Soundness:** 4 excellent
**Presentation:** 4 excellent
**Contribution:** 4 excellent

**Summary:**

The paper examines model update regression in NLP structured prediction tasks. Model update regression is an issue appearing when a model is updated and the new model classifies part of the test examples negatively, while the old model classifies them positively.
The novelty of the paper is that it explores the problem in structured prediction, while it has been previously explored for classification.
The work proposes a new method for correcting the new model from the old one - backward-congruent re-ranking and experimentally shows that it performs better than other previously used methods, such as model ensemble and knowledge distillation.
The work also defines a new measure related to the model update regression problem in structured prediction - negative flip impact.

**Questions:**

n/a

**Strengths And Weaknesses:**

Strengths:
- Studying the model update regression in relation to structured prediction is interesting and important. The contributions are novel enough.
- The paper is very well-written and clear. The problem, previous work, and the proposed solution are clearly described.
- The proposed improvements are meaningful and the experiments show that the proposed method for correction - backward-congruent re-ranking works better than the alternatives on the explored tasks.

Weaknesses:
I did not find any major issues with this work. I am suggesting some things I found that could improve it further.
- Line 120: maybe add a citation for UD 2.2. On the next line only the POS-tagging dataset is cited.
- Line 182: an example could be beneficial for understanding how the global prediction is thus decomposed into a sequence of next-word predictions.
- Line 217: from the description it is not really clear how checkpoint averaging and prediction ensembling differ.
- Table 5: It is not clear what is the difference between rows 5 and 8 (for dropout-p=0.3)

---

> ### Author Response · Authors · 2022-08-01
> **Response to Reviewer uHkD**
>
> Thank you for your careful review and your recognition of our work. We will follow your constructive suggestions for improving our paper.
>
> We will add the citation to UD2.2.
>
> We will give a concrete visualization to illustrate different factorizations when more space is allowed.
>
> Checkpoint averaging averages the parameters of different models and uses the resulting model for prediction. Prediction ensembling averages the prediction scores from different models. We will make it clearer.
>
> Row 5 and 8 are indeed the same. We will omit row 8 for clarity.

---

### Official Review · Reviewer_TTdd · 2022-07-11

**Rating:** 7
**Confidence:** 3
**Soundness:** 2 fair
**Presentation:** 2 fair
**Contribution:** 2 fair

**Summary:**

This work focuses on the update regression problem for structured prediction. The authors explore two standard structured prediction approaches (graph-based and transition-based). They conduct a set of experiments and introduce a noveal approach to mitigate the update regression problem.

**EDIT after discussion with the authors:**
I still disagree with authors, but they made a good point in our discussion, so I upgraded my grades. I strongly recommend the authors to soften some claim made in the paper, refer that in some cases KL div technique for KD can be used and give proper citations about how one could do that.

**Questions:**

- is there any formal motivation for the contribution in section 5.3?
- is there any reason why the previous work listed in my comment above has not been considered to adapt preivous work in the non-structured case to the structured case?

**Limitations:**

nothing to report

**Strengths And Weaknesses:**

**Strenghts:**
- this work focuses on an important and under-studied problem
- experimental results are interesting

**Weaknesses**

Unfortunately, this paper has many weaknesses. First, and importantly, the main contribution of this paper (section 5.3) is not well motivated and the explanation of the contribution itself is really handwavy. Second, explanation of the different baseline approaches is also too handwavy, making the overall paper difficult to understand and not self-contained.

On the motivations: the authors argue that previous approaches cannot be applied to structured prediction. I strongly disagree with this.

- l 227, "the output distributiion of global prediction is often intractable": in the non-projective dependency parsing case, the distribution can be computed using the Matrix Tree Theorem [1, 2, 3]. If we restrict the problem to projective trees, this can be computed via dynamic programming [4, 5]. For the transition based model, approximated approaches have been explored in the litterature [6]
- l. 222, "the instantiations of KD are usually based on a loss function computing the cross-entropy of output distributions [...] however existing methods are no directly applicable to structure prediction tasks": I don't understand why. First, the next sentence is false (see previous point), but also KL divergence between structured distributions has been studied in the litterature. For non-projective trees, see [7, Section 6.3], for methods based on dynamic programming see [8]
- l. 263, "furthermore, it is unclear how to adapt existing sampling methods to solutions that are not formalized as sequence generation": Recent work has considered sampling from the non-projective tree distributions, including sampling without replacement [9, 10]. Moreover, previous work has also considered perturb-and-MAP approaches [11, 12, 13]. Finally, in the case of dynamic programming algorithms, it is well known that it is possible to sample from the associated exponential familly distributions, see e.g. [14]

**Related work**

As suggested by the comment above, the literature is not properly explored or cited by the authors.
There are similar problems in the introduction and related work section. For example:
- l. 54: authors cite Dozan and Manning (2017) for graph-based parsers, whereas the correct citations is more than 10 years older [15]
- l. 21: for transition-based parsers, they cite Ma et al. (2018), better citations would be [16, 17]

[1] Structured Prediction Models via the Matrix-Tree Theorem (Koo et al.)

[2] Probabilistic Models of Nonprojective Dependency Trees (Smith and Smith)

[3] On the complexity of non-projective data-driven dependency parsing (McDonald and Satta)

[4] Semiring parsing (Goodman)

[5] Differentiable Dynamic Programming for Structured Prediction and Attention (Mensch and Blondel)

[6] Globally Normalized Transition-Based Neural Networks (Andor et al.)

[7] Efficient Computation of Expectations under Spanning Tree Distributions (Zmigrod et al.)

[8] First- and Second-Order Expectation Semirings with Applications to Minimum-Risk Training on Translation Forests (Li and Eisner)

[9] Efficient Sampling of Dependency Structures (Zmigrod et al.)

[10] Unbiased and Efficient Sampling of Dependency Trees (Stanojevi)

[11] Perturb-and-MAP random fields: Using discrete optimization to learn and sample from energy models (Papandreou and Yuille)

[12] Differentiable Perturb-and-Parse: Semi-Supervised Parsing with a Structured Variational Autoencoder (Corro and Titov)

[13] Learning Latent Trees with Stochastic Perturbations and Differentiable Dynamic Programming (Corro and Titov)

[14] See section 17.4.5 of  Machine Learning: A Probabilistic Perspective  (Muprhy) for the idea and Latent Template Induction with Gumbel-CRFs (Fu et al.) for application to CRF-like distribution

[15] Non-projective Dependency Parsing using Spanning Tree Algorithms (McDonald et al.)

[16] A Classifier-Based Parser with Linear Run-Time Complexity (Sagae, Lavie)

[17] Algorithms for Deterministic Incremental Dependency Parsing (Nivre)

---

> ### Author Response · Authors · 2022-08-01
> **Response to Reviewer TTdd**
>
> Thank you very much for your review, especially the listed references. We have read all your suggested references carefully.
>
> ***[knowledge distillation]***
> One major concern is on the applicability of knowledge distillation methods on structured prediction tasks. After a thorough investigation of your suggested literature, we find that this is a misunderstanding likely caused by our writing. First of all, by “the output distribution of global prediction is often intractable”, we actually mean “computationally intractable”. We understand that, for some methods (factorizations), the normalized global probability of an output structure is certainly computable. However, the number of possible outputs is exponential in input length. Thus, it is still prohibitively expensive to enumerate all possible outputs in order to compute “the output distribution”. Following this point, it is computationally intractable to compute the KL divergence of two output distributions, which makes the vanilla knowledge distillation impossible. Second, we have carefully checked the mentioned papers [7, 8], [7] is focused on edge-factored, non-projective spanning-tree models and [8] is  for hypergraph-based methods. In other words, those techniques are designed for specific models (factorizations). As emphasized in our paper (Line 52-63, 227-230),  one major challenge for structure prediction tasks is that the prediction can be factorized in many distinct ways. Therefore, such model-specific techniques cannot be  used  for model updates across different factorizations (e.g., heterogeneous model updates such as deepbiaf ⇒stackptr in dependency parsing). Overall, thank you very much for pointing out these issues and we will revise our writing to make the above clear. Last but not least, please note that we did adapt a general and popular knowledge distillation method (sequence-level KD, Kim et al., 2017) as the main baseline.
>
> ***[sampling methods]***
> For your concern on sampling methods. In “it is unclear how to adapt existing sampling methods to solutions that are not formalized as sequence generation”, the “existing sampling methods” actually mean “top-p and top-k sampling” in context. We make it clear. For your suggested sampling methods, they are sophisticatedly designed for specific models (factorizations). In contrast, we propose a simple and general sampling method, dropout-p sampling, for any kind of model as long as it is neural network-based.
>
> ***[older literature]***
> For your suggestion on citing the older literature, we appreciate your pointers and will add them to our paper.
>
> ***[motivations of BCR]***
> The motivations of BCR is given in Line 246-250: “​​Our motivations are two-fold. First, structured prediction models can produce a number of possible predictions. Second, different predictions may achieve similar error rates but differ by the mistakes they made. Among them, the most likely one according to the old model should have the least prediction flips. These make re-ranking particularly useful for structured prediction.”
>
> ***[adaptability of previous work]***
> For the question on the adaptability of previous work,  we already answered it in the beginning of our response.

---

> > ### Comment · Reviewer_TTdd · 2022-08-03
> > **Response**
> >
> > Dear authors,
> >
> > you wrote: "Following this point, it is computationally intractable to compute the KL divergence of two output distributions, which makes the vanilla knowledge distillation impossible"
> >
> > In the paper, you wrote that one of your model is a biaffine model for the syntactic parser. In this case, you can compute the KL distribution, as explained in [7].
> >
> > The second dataset you use for semantic parsing, TOP, contains as far as I know structures that looks like constituency trees. Then, assuming, I agree, that score decomposes into constituents scores, you can use semirings presented in [8] to compute the KL distribution.
> >
> > So, there is no reason why this is not feasible, both for one model that you have in the paper and, in general, could be done for the 2 datasets you use in the paper. Of course, I agree with you that this requires some sort of decomposition in the output distribution, but this is a very standard assumption in structured prediction. The fact that you don't have this KL baseline for the two dataset is preventing you to compare your work with previous work.
> >
> > So, correct me if I am wrong, but **I really don't understand why you don't have this KL baseline to compare your approach to previous work**.

---

> > > ### Author Response · Authors · 2022-08-04
> > > **On knowledge distillation**
> > >
> > > Thank you very much for your reply and bringing up this issue.
> > >
> > > **1. The computability of the output distribution is not only related to the problem but also related to the methods (factorizations).**
> > > For example, dependency parsing can be edge-factored (biaffine parser) or action-factored (stack-pointer parser). They are totally different factorizations to the same problem.
> > >
> > > **2. Under certain factorizations, the KL divergence between two output distributions can be tractable. However, for other factorizations, it is not.**
> > > We acknowledge that some techniques have been proposed to approximate the KL divergence (e.g., [7, 8]). However, they are designed for specific factorizations (e.g., [7] is only for edge-factored parsers, [8] is only for semiring-based parsers). For other factorizations such as the seq2seq parser for semantic parsing, the exact output distribution is computationally intractable (the number of possible outputs is infinite). For the stack-pointer parser, it is computationally expensive to compute the output probabilities of all possible outputs (there are an exponential number of entries and we need every one of them).
> > >
> > > **3. The goal of this paper is to study the regression problem between mode updates. In practical view, we cannot assume the models follow certain factorizations. Also, we cannot assume the old and new models follow the same factorization.**
> > > Therefore, those model-specific techniques have very limited application scenarios (e.g, [7] can only be used for two edge-factored parsers). Particularly, they cannot be used for model updates across different factorizations. We are looking for general solutions that are model-agnostic (e.g., the Sequence-level Knowledge Distillation baseline we adapted and the Backward-Congruent Re-ranking we proposed).
> > >
> > > **4. We have already implemented another very popular and generic knowledge distillation baseline (sequence-level knowledge distillation, Kim et al, 2017).**
> > > Kim et al, (2017) proposed a simple and effective knowledge distillation variant. It has been widely used for structured prediction tasks. In this work, we adapted it to tackle the regression problem. We believe that developing more effective knowledge distillation is a promising research direction, but is not the focus of this paper.
> > >
> > > To make our claim clear to readers, we will soften the sentence "existing methods are no directly applicable to structure prediction tasks" and add more discussion in Sec. 5.2 about your suggested references. Thank you vey much for pointing out these missing references.
> > >
> > > Hope the above address your concerns. Let us know if you have further questions!

---

> ### Comment · Reviewer_TTdd · 2022-08-09
> **Main review edit**
>
> After discussion with the authors, I changed my grade, see the summary section.

---

### Author Response · Authors · 2022-08-03
**General Response (edited for AC and SAC after author-reviewer discussion)**

Thank you for the comprehensive reviews and thoughtful comments.

We are excited with the recognition that "this work focuses on an important and under-studied problem"(Reviewer TTdd) and "the writing is good and clear"(Reviewer XYXe), "the exploration of "new" models is quite thorough"(Reviewer EpNN) and "the proposed improvements are meaningful" (Reviewer uHkD). We are thrilled with the reviewers' acknowledgment that "The BCR technique is simple and I could see it being widely adopted for this task", "The dropout-p sampling technique is interesting and a novel way of getting diverse samples", and "The problem, previous work, and the proposed solution are clearly described.".

Some reviewers (TTdd and XYXe) mainly have questions on the adaptability of knowledge distillation methods. In the context of model update in structured prediction, most existing methods cannot be directly applied for two reasons. First, the output distribution is often computationally intractable due to that the number of possible outputs grows exponentially in input length. Second, knowledge distillation across different factorizations (e.g., heterogeneous model update) can be very difficult since most model-specific methods cannot be applied. We adapted a popular and generic knowledge distillation method (sequence-level knowledge distillation, Kim et al, 2017) as one main baseline. More importantly, We proposed a simple and general approach, BCR, which achieves better performance than the knowledge distillation baseline. **Regarding this issue, we are grateful for the through and insightful discussion with Reviewer TTdd and happy to see we finally reach a consensus. We will improve our paper following the reviewer's suggestions.**

Reviewer XYXe pointed out that we only studied syntactic parsing. However, the fact is that we studied both syntactic parsing and semantic parsing. Both two are graph-structured prediction tasks. We argue that graph-structured prediction plays a considerably large part in the whole NLP community. **Regarding this issue, unfortunately, Reviewer  XYXe did not respond during the Author-Reviewer discussion period.**

Below, we respond to each reviewer separately. Please let us know if you have additional questions or comments!

---

> ### Comment · Reviewer_TTdd · 2022-08-03
> **Response**
>
> You wrote: "the output distribution is often computationally intractable due to that the number of possible outputs grows exponentially in input length"
>
> But I strongly disagree with this point: for the two dataset you use, this would have been trivial to implement (via the Matrix Tree Theorem for dependency parsing and via dynamic programming for the semantic parsing dataset). These are standard algorithms in NLP and structured prediction. I don't understand why you don't have this baseline to compare your approach with previous work.

---

> > ### Author Response · Authors · 2022-08-04
> > **On Knowledge Distillation (KL divergence)**
> >
> > Thank you very much for your reply and bringing up this issue.
> >
> > **1. The computability of the output distribution is not only related to the problem but also related to the methods (factorizations).**
> > For example, dependency parsing can be edge-factored (biaffine parser) or action-factored (stack-pointer parser). They are totally different factorizations to the same problem.
> >
> > **2. Under certain factorizations, the KL divergence between two output distributions can be tractable. However, for other factorizations, it is not.**
> > We acknowledge that some techniques have been proposed to approximate the KL divergence (e.g., [7]). However, they are designed for specific factorizations (e.g., [7] is only for edge-factored parsers). For other factorizations such as the seq2seq parser for semantic parsing, the exact output distribution is computationally intractable (the number of possible outputs is infinite).
> >
> > **3. The goal of this paper is to study the regression problem between mode updates. In practical view, we cannot assume the models follow certain factorizations. Also, we cannot assume the old and new models follow the same factorization.**
> > Therefore, those model-specific techniques have very limited application scenarios (e.g, [7] can only be used for two edge-factored parsers). Particularly, they cannot be used for model updates across different factorizations. We are looking for general solutions that are model-agnostic (e.g., the Sequence-level Knowledge Distillation baseline we adapted and the Backward-Congruent Re-ranking we proposed).
> >
> > **4. We have already implemented another very popular and generic knowledge distillation baseline (sequence-level knowledge distillation, Kim et al, 2017).**
> > Kim et al, (2017) proposed a simple and effective knowledge distillation variant. It has been widely used for structured prediction task. In this work, we adapted it to tackle the regression problem. We believe that developing more effective knowledge distillation (e.g., using [7,8] )is a promising research direction, but is not the focus of this paper.
> >
> > To make our claim clear to readers, we will soften the sentence "existing methods are no directly applicable to structure prediction tasks" and add more discussion in Sec. 5.2 about your suggested references. Thank you vey much for pointing out these missing references.
> >
> > Hope the above address your concerns. Let us know if you have further questions!

---

> > > ### Comment · Reviewer_TTdd · 2022-08-04
> > > **Answer**
> > >
> > > I think you don't understand my point.
> > >
> > > You use two datasets where you could implement the KL baseline (for syntactic parsing you even have the biaffine model already implemented).
> > >
> > > If you would have focused on dataset where there is no tractable algorithm even with indepedence assumptions (e.g. AMR [1] or SDP if you force the DAG constraint [2]), I would understand why you don't have this as a baseline and I would understand your motivations. But in the case of the two datasets you use, not reporting these KL baselines is very strange IMHO. You are basically refusing to report results using the most standard model and baseline approaches.
> > >
> > > [1] https://amr.isi.edu/
> > > [2] http://sdp.delph-in.net/

---

> > > > ### Author Response · Authors · 2022-08-05
> > > > **Why we do not have the suggested baseline**
> > > >
> > > > We thank the reviewer for his/her valuable time. But we respectfully disagree that we are missing "the most standard" baseline.
> > > >
> > > > **The suggested baseline [7] is model(factorization)-specific and thus has very limited application scenarios**. Concretely, [7] can only be used for model updates between two edge-factored parsers (e.g., two biaffine parsers). For dependency parsing, it cannot be used for model updates between two stack-pointer parsers or between a biaffine parser and a stack-pointer parser. Specifically, Even if we add [7] as an additional baseline, **it can only be used in **1 out of 6** experiment setups in our paper**. As explained in our earlier response, **for the problem of model update regression, from a practical view, we should not assume the models follow certain factorizations**.  Moreover, we should not assume the old model and the new model follow the same factorization. Therefore,  **we strive to develop generic solutions that do NOT depend on any specific factorization**.
> > > >
> > > > Another important thing is that **we have chosen a very strong alternative for knowledge distillation (i.e., sequence-level knowledge distillation, Kim et al, 2017)**. Most importantly, sequence-level knowledge distillation is **model-agnostic and thus is well-suited to the problem of model update regression**. In fact, Kim et al, (2017) is a *de facto* choice for structured prediction tasks. It has already been cited **679** times. In contrast, the suggested [7] was published in 2020 and has only **2** citations. **We really cannot agree that [7] is "the most standard" approach.**
> > > >
> > > > Anyway, we thank the reviewer for pointing out the reference. We are happy to add related discussions to our paper. However, we respectfully disagree that missing the baseline affects the major contribution of this paper.

---

> > ### Author Response · Authors · 2022-08-09
> > **Factual Errors in Reviewer TTdd's Understanding on Existing Techniques**
> >
> > We really appreciate the reviewer for his/her hard work. However, we would like to raise some issues in the feedback from Reviewer TTdd. We kindly request special attention to them during the discussion period.  Thank you very much!
> >
> > **Reviewer TTdd insists that the output distribution of dependency parsing and conversational semantic parsing is computationally tractable**. The reviewer points to Matrix Tree Theorem for dependency parsing and dynamic programming for semantic parsing. However, after our through investigation, we find that the reviewer's opinion is incorrect. The references suggested by the reviewer only addressed the computation of the partition function and the marginals but did not address the output distribution. **In fact, one can prove that the output distribution is computationally intractable**. See below.
> >
> > For dependency parsing and conversational semantic parsing, the number of possible outputs is exponential in input length. => The output distribution has an exponential number of terms and we need each and every one of them. => this means the least time complexity for computing the exact output distribution is exponential. => In other words, the problem is intrinsically NP-hard.
> >
> > We acknowledge that some methods [e.g., 7] may approximate the KL divergence of two output distributions (without computing the exact output distributions). However, these methods are model-specific and thus are not generic enough for our problem (i.e., addressing the regression problem between arbitrary models). The limitations of these methods are detailed in the other thread ([On Knowledge Distillation (KL divergence)](https://openreview.net/forum?id=4cdxptfCCg&noteId=6GgxvORJXrF3)).

---

> > > ### Comment · Reviewer_TTdd · 2022-08-09
> > > **What**
> > >
> > > I don't understand this comment.
> > >
> > > For dependency parsing, if you assume arc score decomposition, you can **EXACTLY** compute the partition function using the matrix tree theorem and arc marginals via backpropagation. This is really standard and an implementation can be found here: https://github.com/harvardnlp/pytorch-struct/blob/master/torch_struct/deptree.py#L223
> > >
> > > Then, for knowledge distillation, one can use a loss function between the "old" and "new" models based on the KL divergence between the two distribution over dendency trees, see for example equation 8 in this paper you cite: https://arxiv.org/pdf/2011.09161.pdf
> > > This KL divergence can be computed **EXACTLY**, see section 6.3 in this paper I already refered to: https://arxiv.org/pdf/2008.12988.pdf
> > >
> > > For semantic parsing, to the best of my knowledge, the structure it contains are CFG-like structures. Then, the KL divergence in the loss can be computed **EXACTLY**, again, as explained on page 9, paragraph "Cross-Entropy and KL Divergence" of this paper: https://www.cs.jhu.edu/~jason/papers/li+eisner.emnlp09.pdf
> > > (the term "approximation" does not refer to the KL divergence in this paragrapgh but what computes the automates in another paper they cite)
> > >
> > >
> > > Obviously, this is true only if the distrributions factorize nicely. But this factorization is very standard + often gives SOTA results with neural networks (one obvious counter example is machine translation).
> > >
> > > My argument is as follows: authors could implement this baseline for the biaffine parser where the score decomposition allows to use the matrix tree theorem. This would allow the authors to maybe have an argument like "our approach works as good as the standard KL div approach" for this problem, with the extra benefit that their approach could also be used for distributions where it is not possible to compute the KL div.

---

> > > > ### Author Response · Authors · 2022-08-09
> > > > **response**
> > > >
> > > > thanks for the reply.
> > > >
> > > > For dependency parsing, yes, we agree that the matrix tree theorem "addressed the computation of the partition function and the marginals" but "it did not address the output distribution". The number of dependency trees is exponentially large! (You can compute the total sum of the probabilities of some dependency trees efficiently, but for the whole output distribution, we need the output probability of every and each possible tree ) **We really do not understand how to get the exact output distribution.** Your suggested implementation only computes the marginals. **Let us be clear: the output distribution is computationally intractable.**
> > > >
> > > > In fact, your suggested reference (Section 6.3 in [7]) partly support the above claim, quoted ***"To the best of our knowledge, no algorithms to compute the Kullback–Leibler (KL) divergence between two graph-based parsers (nor its gradient) have been given in the literature. We show how this can be achieved easily within our framework".*** Note that this paper was published in 2020. if one could simply compute the output distributions then compute the KL distance, how would this paper make such a claim?
> > > >
> > > > Lastly, these methods for computing KL divergence assume specific factorizations. Concretely, [7] can only be used for model updates between two edge-factored parsers (e.g., two biaffine parsers). For dependency parsing, it cannot be used for model updates between two stack-pointer parsers or between a biaffine parser and a stack-pointer parser. Therefore, **even just for dependency parsing, it can only be used in 1 out of 3 experiment setups in our paper**. As explained in our earlier response, for the problem of model update regression, from a practical view, **we should not assume the models follow certain factorizations**. Moreover, we should not assume the old model and the new model follow the same factorization. Therefore, **we strive to develop generic solutions that do NOT depend on any specific factorization**.
> > > >
> > > > The reviewer wrote ```Obviously, this is true only if the distrributions factorize nicely. But this factorization is very standard + often gives SOTA results with neural networks```. Yes, edge-factored parsers are popular choices but transition-based and seq2seq based parsers are also popular. The focus of this paper is not to develop a strong parser but to reduce the regression problems when updating between them.

---

> > > > > ### Comment · Reviewer_TTdd · 2022-08-09
> > > > > **Answer**
> > > > >
> > > > > I don't get what you mean by "Let us be clear: the output distribution is computationally intractable". Enumerating all possible outputs and computing their probabiliity? Why would you do that? What you need to compute is the KL divergence, which can be feasible if the two model factorizes similarly (e.g. 2 biaffine parsers). You obviously don't want to enumerate explicitly all possible outputs, but use the underlying decomposition to make the computation tractable.
> > > > >
> > > > > The statement in [7] "To the best of our knowledge, no algorithms to compute the Kullback–Leibler (KL) divergence between two graph-based parsers" is a bit overexaggerated. The matrix tree theorem is older than 2020, and the decomposition that allows to compute the KL divergence is well known in the exponential familly litterature. But it is probably less known in other part of ML and NLP communities.
> > > > >
> > > > > I still think that it is a loss to not have this KL based KD as a baseline (for both dataset, i.e. using a graph based dependency parser for syntactic dep parsing and a span based parser for semantic parsing). It would be a great addition to the paper and, if your proposed method compares favorably or equivalently when used between 2 factorized models, would make the paper way stronger).
> > > > >
> > > > > If not, I think you should still make it clear in the paper that losses like equation 8 in this paper you cite: https://arxiv.org/pdf/2011.09161.pdf could be used in some cases, even for structured prediction.

---

> > > > > > ### Author Response · Authors · 2022-08-09
> > > > > > **we think we finally understand each other**
> > > > > >
> > > > > > Thank you very much for your time!
> > > > > >
> > > > > > Yes. We agree that some methods can compute the KL divergence of two output distributions **without computing the exact output distributions**. But the exact output distribution is still intractable (we mean it is prohibitively expensive to enumerate all possible outputs).
> > > > > >
> > > > > > We really appreciate your pointers to these methods. Although we argue that these methods are model-specific but we strive to develop model-agnostic solutions for the nature of model update regression, we acknowledge they are indeed worth discussing. Also, the discussion is very helpful for making our contributions clear to readers, we will revise the current writing and add elaborate discussion in Sec. 5.2 about your suggested references. We agree that adding these methods as additional baselines can only strengthen our paper and will run additional experiments. Thank you very much for pointing out these missing references!
> > > > > >
> > > > > > Thank you very much for your active discussions with us.

---

### Meta-Review · Area_Chair_EPpd · 2022-08-29

**Recommendation:** Accept
**Confidence:** Certain

**Metareview:**

This is a good paper with a topic that is very important in practical scenarios but does not have many off the shelf solutions, and I find that this paper makes an attempt to this end.  In addition, I am happy to see the discussion with the reviewers, most of whom suggest acceptance.

**Award:**

No

---

### Decision · Program_Chairs · 2022-09-14

Accept